# OpenReview forum: "MAAD Private: Multi-Attribute Adversarial Debiasing with Differential Privacy"
_ICLR.cc/2024/Conference — Submitted to ICLR 2024_

### Official Review · Reviewer_DNiP · 2023-10-30

**Soundness:** 2 fair
**Presentation:** 2 fair
**Contribution:** 1 poor
**Rating:** 3
**Confidence:** 4

**Summary:**

This paper considers the problem of developing ML models while considering all utility, fairness and privacy concurrently.
This paper takes an in-processing approach i.e., occurs during training as opposed to pre-processing or post-processing approaches.  The proposed method consists of a “predictor” and an “adversary” working as follows. The predictor model estimates the target label from non-sensitive features. The adversary model estimates the sensitive attribute based on the predictor's output. This paper propose a loss function that adjusts the predictor model such that it learns the primary task well without dependence on sensitive attributes.

**Strengths:**

1- Studying an important problem of developing ML models while considering all utility, fairness involving multiple sensitive attributes and privacy concurrently
2- Experiments using two tabular datasets (Adult and COMPAS) with two sensitive attributes (race and sex) and a natural language dataset  with 4 sensitive attributes (age, gender, ethnicity, and country)

**Weaknesses:**

1- The contributions of this paper are limited as the proposed method is a straightforward combination of three existing works:
1) Adversarial debiasing  Zhang et al. (2018) that encourages the model to learn domain-invariant features by minimizing the success of an adversarial component that aims to predict sensitive attributes;
2) Fairness algorithms using differential privacy Jagielski et al. (2019) that injects DP into existing fairness algorithms;
3) Kang et al. (2022) that extends adversarial debiasing techniques to multiple sensitive attributes

In particular, this paper combines the above existing methods to i) debias multiple sensitive attributes; and ii) extend the privacy guarantees to all attributes as opposed to only sensitive attributes. I acknowledge and appreciate authors being honest on this front though.

2- Lack of fairness guarantees. It seems that the proposed method does not satisfy any fairness guarantees, it just minimizes the ability of a particular adversary model to predict sensitive attributes. This limitation is inherited from adversarial methods which cannot guarantee protection against other models than the one considered.

3- Inconsistent and surprising results. After looking at Tables 2-4, the performance of DP and DP+bias is not intuitive to me:
1) Table 2 and Table 3 show that DP + Debias method achieves better performance than than DP method. This is surprising.
2) However, Table 4 shows that DP + Debias achieves worse performance than DP method. Although this is expected, this is not consistent with the results of other datasets in Table 2 and Table 3.

4- Shallow discussion of results. I would appreciate it if authors could analyse their results, this could help to shed light on the above #3 concern.

5- Some ambiguity in the algorithm and the proposed method.
1) Is line 10 and 14 in algorithm 1 are done per-example? If not, how this follow the requirements of satisfying DP guarantees?
2) The description of the projection part is missing therefore it is hard to understand why this projection does not break DP guarantees.

6- Experimental choices including hyperparameters and privacy budgets are not justified/studied. For example how sensitive the performance of the proposed method is to the value of the hyperparameter introduced by the proposed loss function.

7- Overclaims. The abstract claims that the proposed method meets various fairness criteria. However, I can see only results of Equality of Odds and I think Section 2.1 confirms my concern as it says "In the current work, we adopt Equality of Odds (Hardt et al., 2016) as a definition of fairness". Section 4.5 also highlights that only Equality of Odds is considered for the evaluation.

8- High computational costs. I would report computational costs to see the effect of the proposed method on the computational costs. If I understand correctly the proposed method computes per-example gradient for both predictor and adversary model, so I would expect an increase in the computational costs.

9- Figure 1 is not referred in the text.

10- Lack of empirical comparisons with existing DP+fairness works.

11- Unclear assumptions. For example, I would highlight/justify that this paper only considers group fairness and not individual fairness and a motivation for it.

**Questions:**

Please see comments in the Weaknesses box especially 1-6.

---

### Official Review · Reviewer_id92 · 2023-10-31

**Soundness:** 2 fair
**Presentation:** 2 fair
**Contribution:** 2 fair
**Rating:** 3
**Confidence:** 4

**Summary:**

The paper proposes a method to guarantee differential privacy (DP) and (approximate) group fairness in learning. The approach is based on adversarial debiasing: while the main model is learning to predict the labels, an adversarial model tries to predict the sensitive attributes given access to the final layer outputs of the main model. The main model is then jointly trained also to minimize the success of the adversarial model, which encourages the learned model to be less dependent on the sensitive features. In the current paper, adversarial debiasing is combined with other techniques: fairness and privacy (from Jagielski et al. 2019) and multi-attribute debiasing (from Kang et al. 2022).

**Strengths:**

* Combining privacy and fairness is a timely topic, in the sense that there has been growing interest on it during the last years.

* Some parts of the paper are clear to read (see weaknesses for some comments on the other parts).

**Weaknesses:**

* There are parts in the paper that could be made clear; most importantly related Alg.1, writing out in equations the projection, clipping and noising, the subtractions (lines 11-12 in Alg.1), etc. As it currently stands, with text description and some math, especially with the details like projections or DP, I have an uneasy feeling that I cannot exactly tell what is going on in the algorithm.

* I have some reservations on the originality of the proposed method: it seems like a fairly obvious combination of existing techniques, that to me does not seem to be any bigger than the sum of its parts.

**Questions:**

1) Sec.5: looking at the results, especially in Table 2, I have a hard time believing that DP+debiasing would have better results than only debiasing. Can you explain?

2) How are the hyperparameters tuned in the experiments?

3) Why are the privacy budgets in Sec.4.4 chosen as they are? While there is a comment on why large epsilons are acceptable, I do not agree with the sentiment that this is generally okay because someone uses even bigger values.

4) Sec. 4.5: provide at least some description of the metrics used.

5) Sec.4.5: can you clarify where you enforce EoD in the proposed method?

6) Contrary to what is stated Sec.6.2 on reproducibility, there does not seem to be any supplementary material provided?

7) Sec.6, Modularity of DP: "This ensures both unbiased learning and individual data protection". Do you actually have some guarantees beyond empirical results showing that debiasing approach used guarantees some given amount of fairness?

8) Please define the neighbourhood relation explicitly in defining DP.


### Smaller comments, typos etc (do not require comments):

i) Sec.3, on Jagielski et al. only noising the Auditor: note that Jagielski et al. assume that the Learner does not have access to the sensitive feature, and since the other features are assumed public, adding DP on those is not necessary. Please fix the claim to clearly state that you have differing settings.

ii) Note that delta in DP is only roughly the maximum catastrophic failure probability.

iii) Sec 2.2 end: moments accountant, and any other privacy accounting method based on Renyi DP, is at least a bit loose in approximate DP space (see e.g. Zhu et al. 2019).

iv) End of Sec.6.1: missing ref?

References:
Yaghini et al. 2023: Learning with Impartiality to Walk on the Pareto Frontier of Fairness, Privacy, and Utility.
Zhu et al. 2019: Optimal Accounting of Differential Privacy via Characteristic Function.

---

### Official Review · Reviewer_5K9B · 2023-10-31

**Soundness:** 2 fair
**Presentation:** 1 poor
**Contribution:** 2 fair
**Rating:** 3
**Confidence:** 5

**Summary:**

This work presents an enhancement to the adversarial debiasing approach, enabling it to account for multiple sensitive attributes while upholding a privacy-conscious learning paradigm. Extensive empirical results are conducted to demonstrate the ability of debiasing while providing privacy protection of the proposed method.

**Strengths:**

- Extensive empirical results on benchmark datasets are conducted to evaluate the models.
- Using well-known metrics for Fairness evaluations

**Weaknesses:**

- Marginal contribution regarding novelty.
- Unclear privacy guarantee since there's no theoretical analysis w.r.t privacy.
- The presentation of the work is not at the level of an academic publication.

**Questions:**

- What is the privacy guarantee?
- Why do you need two terms of L_a in Eq. 2?
- What is the privacy risk incurred from U?

---

### Official Review · Reviewer_VSVe · 2023-10-31

**Soundness:** 2 fair
**Presentation:** 2 fair
**Contribution:** 3 good
**Rating:** 3
**Confidence:** 3

**Summary:**

This paper studies fair classification under the constraint of differential privacy. Existing work on adversarial debiasing, in which an adversary tries to predict a sensitive attribute via gradient reversal, can debias only a single sensitive attribute. This paper proposes a modification to the adversarial debiasing approach which can debias multiple sensitive attributes.

**Strengths:**

The intersection of privacy and fairness is a very interesting and important line of research! Being able to extend fairness from single attributes to multiple sensitive attributes seems like a practical and necessary direction. The authors also seem very knowledgeable on the background and related work.

**Weaknesses:**

I find that the presentation style has some weaknesses…The writing style is very high-level which makes it hard to dig into the technical details. Mathematical notations are introduced informally and I don’t know if I’d be able to implement Algorithm 1 correctly based on the pseudo-code. Furthermore, the privacy guarantee for Algorithm 1 is not stated in the paper.

For an empirical paper, the experiments seem weak. Only three datasets are evaluated; only one privacy regime per dataset is studied; and it is difficult to compare the DP results across datasets because the $\epsilon$ value is different for each dataset. I also don’t see a strong takeaway message from the experimental results and the only baseline is a basic classification model.

**Questions:**

1. How were the different $\epsilon$ values in the experiments chosen?
2. The patterns on the experimental results seem consistent between the two tabular datasets (Adult and COMPAS), but then things take a mysterious left turn at the hate speech dataset. Is there an explanation for why debiasing improves the performance against the non-private baseline for the hate speech dataset, but debiasing + DP hurts the performance against DP-only? And why this would be the opposite for the other two datasets (where debiasing helps in the private setting but hurts in the non-private setting)?

Minor comments:
1. The conclusion alludes to four datasets, but I only count three?
2. Some potential typos:
    1. “Debasing” in the caption of Figure 1
    2. “following advantages: reasons:” in Section 6
    3. undefined reference “??” at the very end of Section 6.1

---

### Official Review · Reviewer_LNLd · 2023-11-05

**Soundness:** 3 good
**Presentation:** 3 good
**Contribution:** 1 poor
**Rating:** 3
**Confidence:** 4

**Summary:**

The paper enhances the adversarial debiasing approach, enabling it to account for multiple sensitive attributes while upholding a privacy-conscious learning paradigm.

**Strengths:**

1. The paper presentation is clear.
2. The paper is well motivated.

**Weaknesses:**

1. The proposed method is incremental and lacks novelty. It is a simple combination of Kang et al. (2022) with DPSGD.
2. No proper baselines or related comparison justification for the paper. There are a number of works that share the same or similar setting as the paper (see the survey at https://arxiv.org/pdf/2305.06969.pdf. However, the paper does not provide a comparison with those methods.
3. Experiments only show the main results without additional in-depth analysis and ablation study. For example, how different parameter changes would affect the performance of utility and fairness trade-offs.

Minor
1. Typos: “??” in the paper.

**Questions:**

Please respond to the weaknesses above.

---

### Meta-Review · Area_Chair_kv9y · 2023-12-06

**Metareview:**

This paper gives an approach for achieving fairness in differentially private (DP) classification models. The approach is to use an adversarial training where the model tries to predict the label while the adversary try (given the predicted label) to predict the sensitive feature to violate a given the fairness criteria (which is chosen to be "Equality of Odds" in this paper). DP is achieved during training via noise addition in a standard "DP-SGD" manner (Abadi et al., 2016). The authors demonstrate through experiments that this gives good fairness guarantee while sometimes also improve upon vanilla DP-SGD training in terms of accuracy.

# Weakness

- Lack of novelty: The adversarial training approach without DP was proposed by Zhang et al. (2018). The paper fails to explain why this is not a straightforward adaptation of the aforementioned previous works of Zhang et al. and Abadi et al.

- Lack of comprehensive experiments: Given that the paper doesn't have any theoretical claims, the experiments are not sufficiently comprehensive. The architecture, privacy parameters, batch size, adversary weight etc. are all fixed (without much justification) and no attempt to study the effect of any choice of parameters is made.

- Lack of clarity in writing: The algorithms are described in rather informal fashion, adding to the confusion even more.

**Justification For Why Not Higher Score:**

Given the apparent lack of novelty of the techniques and the unconvincing writing, this paper is clearly not ready to be published as it stands.

**Justification For Why Not Lower Score:**

N/A

---

### Decision · Program_Chairs · 2024-01-16

Reject